# Analysis of the Compositional Features and Codon Usage Pattern of Genes Involved in Human Autophagy

**DOI:** 10.3390/cells11203203

**Published:** 2022-10-12

**Authors:** Zarnain Jamil, Arif Uddin, Syed Sahajada Mahafujul Alam, Arijit Samanta, Nojood Altwaijry, Mohd Ahmar Rauf, Safdar Ali, Mohd Shahnawaz Khan, Muhammad Nadeem Asghar, Mehboob Hoque

**Affiliations:** 1Applied BioChemistry (ABC) Laboratory, Department of Biological Sciences, Aliah University, Kolkata 700160, India; 2Department of Zoology, Moinul Hoque Choudhury Memorial Science College, Hailakandi 788150, India; 3Department of Biochemistry, College of Sciences, King Saud University, Riyadh 11451, Saudi Arabia; 4Department of Surgery, Miller School of Medicine, University of Miami, Coral Gables, FL 33146, USA; 5Clinical and Applied Genomics (CAG) Laboratory, Department of Biological Sciences, Aliah University, Kolkata 700160, India; 6Department of Medical Biology, University of Québec at Trois-Rivieres, Trois-Rivieres, QC G9A 5H7, Canada

**Keywords:** autophagy-related (*ATG*) genes, codon usage bias, mutation pressure, minimum free energy (mFE), natural selection

## Abstract

Autophagy plays an intricate role in paradigmatic human pathologies such as cancer, and neurodegenerative, cardiovascular, and autoimmune disorders. Autophagy regulation is performed by a set of autophagy-related (*ATG*) genes, first recognized in yeast genome and subsequently identified in other species, including humans. Several other genes have been identified to be involved in the process of autophagy either directly or indirectly. Studying the codon usage bias (CUB) of genes is crucial for understanding their genome biology and molecular evolution. Here, we examined the usage pattern of nucleotide and synonymous codons and the influence of evolutionary forces in genes involved in human autophagy. The coding sequences (CDS) of the protein coding human autophagy genes were retrieved from the NCBI nucleotide database and analyzed using various web tools and software to understand their nucleotide composition and codon usage pattern. The effective number of codons (ENC) in all genes involved in human autophagy ranges between 33.26 and 54.6 with a mean value of 45.05, indicating an overall low CUB. The nucleotide composition analysis of the autophagy genes revealed that the genes were marginally rich in GC content that significantly influenced the codon usage pattern. The relative synonymous codon usage (RSCU) revealed 3 over-represented and 10 under-represented codons. Both natural selection and mutational pressure were the key forces influencing the codon usage pattern of the genes involved in human autophagy.

## 1. Introduction

Autophagy is a highly conserved catabolic process which involves the delivery of certain cytoplasmic molecules and organelles to the lysosomes leading to their degradation. Autophagy is of three major types differing in their mode of cargo delivery to the lysosome: macroautophagy, microautophagy, and chaperone-mediated autophagy [1]. Macroautophagy is the primary autophagic pathway in eukaryotes that involves the bulk degradation of cargo via the double-membrane vesicles called autophagosomes directly into the lysosome. This type of autophagy plays significant roles in cellular physiology; the removal of harmful cargo such as protein aggregates and damaged organelles; adaptation to various metabolic stresses; prevention against genomic damage and numerous diseases such as neurodegeneration, cancer, and also ageing [1,2]. Autophagy is mediated by a set of autophagy-related (*ATG*) genes that are evolutionarily conserved and initially discovered in genetic studies on yeast [3]. Currently, there are several protein-coding *ATG* genes found in humans that perform diverse functions including the extracellular delivery of cargo, extracellular release of intracellular cargo, and communication with various cell-signaling pathways [2]. The entire process of autophagy is carried out in multiple phases such as initiation, nucleation, maturation, fusion, and degradation [4]. Various *ATG* genes such as *ULK1* and *ULK2*, *ATG13*, and *ATG101* are involved in the initiation of autophagy [5]. The nucleation and maturation phase involves *BECLIN1*, *ATG14*, *WIPI1*, *WIPI2, ATG3, ATG5, ATG7, ATG10, ATG12, ATG16L1, ATG16L2,* and *ATG4* genes [3,6,7,8,9,10]. Besides the dedicated *ATG* genes, there are several other genes that are involved in the process of human autophagy such as *AMBRA1, ATF6, BAG1, HIF1A, NRBF2, RUBCN*, and many more.

During the translation process, information is transferred through nucleotide triplet codons, that code for the designated amino acids. The protein sequences are comprised of 20 standard amino acids encoded by 61 codons, and 3 codons provide stop signals during protein synthesis. Except for methionine (M) and tryptophan (W), all amino acids have more than 1 codon representing them, a phenomenon known as the codon degeneracy, and these bunches of codons encoding the same amino acid are known as synonymous codons [11]. Numerous studies in different organisms revealed the non-random usage of these synonymous codons, i.e., at the time of translation of mRNA to protein, some codons are more frequently used than the others. This phenomenon of synonymous codons being used in an uneven fashion is called the codon usage bias (CUB), and the more repeatedly used codons in highly expressed genes are referred to as preferred or optimal codons [12]. The genetic code is preserved in all domains of life, but the pattern of the CUB differs among organisms. It may also differ among different tissues within the same organism and has been recorded among a diverse group of organisms, from prokaryotes to higher eukaryotes [13,14,15]. The two main factors that govern the CUB of an organism are natural selection and mutation pressure [16]. Bias in the pattern of codon usage can be explained by two theories, namely, selectionist and neutral/mutational explanations. One group of scientists claims that selection pressure governs the CUB as it supports the potency and efficiency of protein, although the other group claim that mutational bias causes the CUB in genes [17,18].

Several reviews indicated that CUB is also governed by additional factors such as nucleotide composition, gene length, GC content, genetic drift, population size, environmental stress, protein structure, gene expression level, evolutionary age of the gene, DNA replication, RNA structure, codon location, and properties of proteins such as hydropathicity and aromaticity [12,16,19]. The coding sequences (CDS) of the genome are the representation of different gene products and a genome-wide study of the pattern of CUB contributes valuable details on the molecular mechanism and modification of genes; detection of a new gene; and studies based on phylogeny, designing primers, and transgenes, estimating the role and expression level of genes [20,21,22]. Earlier, synonymous mutations in CDS were believed to be “silent” mutations; however, advanced study on the CUB suggests that it might relate to numerous cellular processes and could influence human disorders [23].

In a recent study, the CUB of the *ATG13* gene across eukaryotes has been reported [24]. However, the genetic pattern of the codon usage bias involved in human autophagy is not yet known. Moreover, this remains unclear as to whether all the genes involved in human autophagy (called ‘autophagy genes’ here) display similar codon usage patterns and the factors involved. Elucidating the patterns of codon usage in human autophagy genes would provide novel insights into their genetic characteristics, the molecular evolution of these genes, and contribute to molecular details of the autophagic pathway in humans. Therefore, in our current study, we investigated the composition of nucleotides and pattern of CUB in 224 protein-coding human autophagy genes. Several parameters such as compositional properties, relative synonymous codon usage (RSCU), parity plot, neutrality plot, correlation, and regression analysis between different compositional properties, codon context, correspondence analysis, protein properties, and free energy of mRNA have been studied to recognize the patterns of CUB and the components that govern the codon usage pattern of the human autophagy genes.

## 2. Materials and Methods

### 2.1. Sequence Retrieval

All the protein coding genes involved in the human autophagy process and their accession numbers were retrieved from a curated database, the Human Autophagy Database (HADb) (http://autophagy.lu/) accessed on 7 August 2022 [25]. The resulting genes were verified for their respective function in the human autophagy process by a relevant literature search [26]. The genes were then further confirmed in the HUGO Gene Nomenclature Committee (HGNC) database, and finally a total of 226 protein-coding human autophagy genes were selected for this study (Appendix A). The transcript detail given in the HGNC database for each gene was considered for our analysis. The CDS of 224 human *ATG* genes were extracted from the nucleotide database of the National Centre for Biotechnology Information (NCBI) (http://www.ncbi.nlm.nih.gov) in FASTA format on 8 August 2022 [27]. Exclusively, those sequences were included that contained proper initiation and termination codons, devoid of any internal stop codons, and were the exact multiples of three nucleotides (Appendix A). The proper CDS of two genes (*FAM215A* and *GABARAPL3*) were not found in the sequence database.

### 2.2. Nucleotide Composition

Various compositional properties of human autophagy genes were calculated that included the overall frequency of adenine (A), cytosine (C), thymine (T), guanine (G), and the frequency of nucleotides at the third codon position such as A3, C3, T3, and G3. The overall GC content and GC content at the first (GC1), second (GC2), and third (GC3) codon position were also estimated.

### 2.3. Relative Synonymous Codon Usage (RSCU)

The RSCU is said to be the fraction of observed frequency of a particular codon to its expected frequency under uniform synonymous codon usage [28]. Mathematically, RSCU is represented as,
$$\mathrm{RSCU}=\frac{\;\mathrm{Xij}}{\frac{1}{\mathrm{ni}\;}\sum_{j=1}^{\mathrm{ni}}\mathrm{Xij}}$$
where x_ij_ is the frequency of incidence of j^th^ codon for i^th^ amino acid (any x_ij_ with a value of zero is arbitrarily granted a value of 0.5) and n_i_ is the sum of synonymous codons that encode for the i^th^ amino acid.

### 2.4. Effective Number of Codons (ENC)

The ENC is an index significantly used to determine the extent of CUB [29]. ENC values of human autophagy genes were estimated using the online tool available at http://agnigarh.tezu.ernet.in/~ssankar/cub.php on 8 August 2022 [30,31]. The ENC is computed as:$$\mathrm{ENC}=2+\frac{9}{F_{2}}+\frac{1}{F_{3}}+\frac{5}{F_{4}}+\frac{3}{F_{6}}$$
where F_k_ (k = 2,3,4 or 6) is the mean of F_k_ values of k-fold degenerate amino acids. The F value indicates the probability of two random codons identical for an amino acid.

### 2.5. Parity Plot

The plot of parity rule 2 bias was prepared utilizing the bias of AT [A3/(A3+T3)] plotted on the Y-axis and the bias of GC [G3/(G3+C3)] plotted on the X-axis, maintaining 0.5 as the center of the plot that designates the complementary nucleotides exhibiting no bias [32]. The parity plot analysis was conducted to assess the influence of two principle forces, that is, the effect of mutation pressure and that of natural selection on CUB [33].

### 2.6. Neutrality Plot

The neutrality plot was prepared by plotting the mean GC content at first and second position (GC12) along the Y-axis versus GC3 along the X-axis to explore the mutation-selection equilibrium in the CUB of human autophagy genes. The slope of the regression line estimates the magnitude of mutation and natural selection [34]. The slope nearer to zero or the absence of correlation between GC12 and GC3 specifies the influence of natural selection and the slope closer to 1 indicates the influence of mutation pressure on the CUB of genes.

### 2.7. Correspondence Analysis

The correspondence analysis (CoA), a multidimensional statistical approach, was conducted to evaluate the major tendency of data variation in codon usage patterns amid autophagy genes by placing them in continuous axes [35,36]. Depending on the 59 sense codons exhibiting a particular RSCU value, all the human autophagy genes were plotted in 59-dimensional hyperspace. The CoA for these genes was performed using “Past” software and the pictorial view of the CoA plot was obtained [37].

### 2.8. Codon Context Analysis

An analysis of the codon context was accomplished using the software ANACONDA V2.0 with all the 64 codons along both the axes to identify the codon context patterns of all human autophagy genes [38].

### 2.9. Protein Properties

The physicochemical features of proteins such as polypeptide length, aromaticity, and hydrophobicity might affect the CUB of the encoding genes. The aromaticity (Aromo) scores for all the human autophagy genes that indicate the occurrence of aromatic amino acids (W, Y, F) in the respectively encoded proteins were evaluated by employing a software named CodonW (ver 1.4.2) [39]. The grand average of hydropathicity (GRAVY) score is the measure of hydropathicity or hydrophobicity of a protein. It is assessed by the addition of the product of abundance of an individual amino acid present in a protein and its corresponding index of hydropathy [40]. Its value ranges from +2 indicating the hydrophobicity of a protein to −2 indicating protein hydrophilicity. The GRAVY score was determined utilizing the GRAVY Calculator (http://www.gravy-calculator.de/) accessed on 20 August 2022 [41]. The isoelectric points (pI) of the proteins were calculated using web server IPC 2.0 (www.ipc2-isoelectric-point.org) accessed on 20 August 2022 [42,43].

### 2.10. Minimum Free Energy (mFE) of mRNA

The mFE values of the autophagy gene mRNAs were calculated by using the web server named RNAfold available at http://rna.tbi.univie.ac.at/ accessed on 20 August 2022 [44,45]. The complete sequences of mRNAs are provided in Appendix A. The obtained negative mFE values in kcal/mol were indicative of the loss of energy during stabilization of the mRNA molecules. For the statistical analysis, the absolute values of mFE were used. The higher the absolute values of mFE, the greater the loss of energy and the higher the stability of the conformation attained by mRNA.

### 2.11. Software

The nucleotide compositions and RSCU values of all the human autophagy genes were quantified with the aid of CAIcal server (http://genomes.urv.es/CAIcal) accessed on 8 August 2022 [46,47]. ENC scores were evaluated by the virtual tool available at http://agnigarh.tezu.ernet.in/~ssankar/cub.php# [30,31]. All the GRAVY scores were computed accessing the GRAVY Calculator available online (http://www.gravy-calculator.de/) accessed on 29 September 2022 [41]. A codon context analysis was performed using Anaconda software. The RSCU heatmap was generated using the HemI 2.0 software [48]. Aromo values were obtained by codonW software. The CoA was accomplished using the Past software. An analysis of correlation was carried out to study the possible relation among variables using Karl Pearson’s product moment method. Microsoft Excel 2013 was employed to perform the majority of the statistical analyses used in our study.

## 3. Results

### 3.1. Human Autophagy Genes Contain Higher GC Content

The nucleotide compositions of genes provide a vivid illustration regarding the arrangement of codon usage. In this study, the complete nucleotide composition of CDS and composition of nucleotide of third codon positions of human autophagy genes were calculated (Figure 1). The mean of G% was recorded to be the highest (26.59%), which was followed by A% (25.91%), C% (25.65%), and T% (21.85%) (Figure 1A). These genes were observed to be rich in GC bases with the overall GC content of 52.23%, while the AT content was found to be 47.77%. At the third position of codon, the mean C3% was recorded to be the maximum (30.48%) followed by G3% (28.95%), T3% (21.85%), and A3% (18.72%) (Figure 1B). The GC content analysis at various codon positions revealed that the highest GC content occurs at the third codon position (59.43%), and the lowest at the second position (41.44%), while GC1% was found to be transitional (55.84%) between the two (Figure 1C).

### 3.2. Human Autophagy Genes Show Low Codon Usage Bias (CUB)

The ENC values of human autophagy genes were determined to quantify the extent of variation in synonymous codon usage. The extent of deflection patterns of codon usage within a gene as compared to the uniform utilization of synonymous codons is evaluated by the ENC, which are independent of the gene and polypeptide length. The ENC value is inversely correlated with the CUB and it ranges from 20 to 61 indicating extreme and no bias, respectively [29]. The value of ENC ≤ 35 is generally considered as significant CUB [29,49]. In our study, the ENC values of genes involved in human autophagy were found to be within a range between 34.14 and 54.6 with an average of 45.05, suggesting overall a low CUB (Appendix A).

### 3.3. RSCU Analysis Indicates Stable Genetic Composition of Human Autophagy Genes

Uniform uses of all synonymous codons are not generally observed, and some are more often used than their synonymous counterparts. RSCU values of individual synonymous codons were estimated to obtain knowledge about the abundance of specific codons. RSCU values are generally categorized into four groups: (a) over-represented with RSCU values above 1.6; (b) under-represented with RSCU values below 0.6; (c) more frequently used with RSCU between 1 and 1.6; and (d) less frequently used with RSCU between 0.6 and 1 [50,51]. In our analysis, most of the codons had their mean RSCU values in between 0.6 and 1.6, which indicated that the genetic composition of the genes was stable with 26 more frequently used, and 20 less frequently used codons (Figure 2). There was a total of 10 under-represented codons with RSCU values < 0.6 (CGT, ACG, CAA, GTA, CCG, TTA, GCG, ATA, CTA and TCG), and only three over-represented with RSCU > 1.6 (CTG, GTG and ATC). The comprehensive RSCU values for the entire number of codons, that is 59, contributing to codon degeneracy in genes involved in human autophagy is represented as a heatmap in Appendix A.

### 3.4. Base Composition Influences the Codon Usage in Human Autophagy Genes

To explore the effect of codon usage patterns in the human autophagy genes in terms of composition of various nucleotides, scattered regression plots were created between the ENC and different compositional properties such as the overall nucleotide contents (A, T, G, and C), constituent nucleotides present at the third position of codons (A3, T3, G3, and C3), and all of the GC contents (GC, GC1, GC2, and GC3) (Figure 3A-L). The regression coefficients of ENC on A, T, A3, and T3 were found to be positive (0.594, 0.813, 0.428, and 0.477, respectively) and negative on G, C, G3, C3, GC, GC1, GC2, and GC3 (−0.879, −0.592, −0.524, −0.351, −0.430, −0.373, −0.291, and −0.244, respectively). Since the ENC is observed to be a non-directional estimation of the CUB, the higher values of the ENC indicate low CUB and vice-versa [50]. Therefore, the negative regression coefficients of ENC on G, C, G3, C3, GC, GC1, GC2, and GC3 indicate their positive influence on the CUB.

Moreover, the correlation analysis was performed between the ENC versus different compositional attributes to study the consequence of nucleotide constituents on patterns of codon usage (Figure 3 A–L insets). A significantly higher value (*p* < 0.01) of correlation that was found to be positive was established between the ENC and A, T, A3, and T3, while some of the correlation values were observed to be significantly (*p* < 0.01) negative as between the ENC and G, C, G3, C3, and nearly all of the GC constraints such as GC, GC1, GC2, and GC3. Further, Figure 3M shows the correlation analysis between codon usage and the GC3 bias of human autophagy genes depicting their interrelationship. It was found that the codon usage of nearly all the codons that ended with nucleotide GC were correlated in a positive manner with GC3 (except the codons AGG and TTG), and conversely, all the codons that ended with nucleotide AT were found to be correlated with GC3 in a negative manner. These results indicated that with an increase in GC bias, the codons ending with GC displayed increased usage of codons, while the codons ending with AT displayed decreased usage. This implied that the bias of codon usage in human autophagy genes was influenced by GC ending codons.

### 3.5. Both Natural Selection and Mutation Pressure Determine the Codon Usage in Human Autophagy Genes

#### 3.5.1. Parity Plot Analysis

The parity rule 2 bias plot analysis of [A3/(A3+T3)] vs. [G3/(G3+C3)] was conducted to comprehend the relationship among A/T and G/C constituents which was useful for estimating the relative quantity of natural selection and mutational pressure upon the constituent of genes. In the event that mutation pressure acts on the CUB of genes, both nucleotides, namely, AT as well as GC should be equivalently dispersed among the degenerate group of codons. On the contrary, a disproportionate use of AT and GC suggests that both forces such as mutational and selection pressure influence the codon usage bias [33]. In this study, the mean GC bias was found to be 0.4948 ± 0.0602 and the mean AT bias was 0.4545 ± 0.0622. The observed asymmetrical distribution of the GC and AT indicated that both pressures (mutational and natural selection) affected the CUB genes involved in human autophagy (Figure 4A). Further, the residual analysis of the parity plot showed a random distribution of the points over the straight horizontal line that demonstrated the regression model’s linearity (Figure 4B).

#### 3.5.2. Neutrality Plot Analysis

The decisiveness of the CUB of genes is attributed to the two main forces of evolution: mutation and natural selection. The neutrality plot was produced to investigate the contribution of these two factors by plotting the mean of the GC of the first and second positions of codons (GC12) on GC3. The coefficient of regression nearer to 0 specifies natural selection and the regression coefficient closer to 1 indicates the mutation pressure that helps in framing the CUB [34]. In the neutrality plot of all genes involved in autophagy in humans as shown in Figure 4C, the slope of the regression line that illustrates the regression coefficient was found to be 0.213, suggesting the influence of natural selection on the CUB of genes. On the other hand, the plot also depicted that the points were diagonally distributed, suggesting the impact of mutation pressure on codon bias in human autophagy genes. Moreover, the codons GC12 and GC3 exhibited a moderately strong and positive correlation among them (r = 0.588, * *p* < 0.01), suggesting the action of directional mutation on the entire positions of codons. We further noticed a broad scale of dispersal of GC3 values that indicates mutation pressure controlling the codon usage bias of autophagy genes. Overall, these results showed that both of the dominating forces of natural selection and mutation pressure could be controlling the CUB of genes related with human autophagy. Moreover, the residual analysis of the neutrality plot showed the random distribution of the scattered points across the horizontal line, suggesting the regression model’s linearity among the codons GC12 and GC3 (Figure 4D).

#### 3.5.3. Correlation Analysis of the Third Codon Position against Base Composition

A correlation analysis among complete nucleotide composition and the composition of nucleotides at the third position of codon was performed to evaluate the effect of the evolutionary forces on the CUB. We noticed highly significant correlation values that were observed to be positive as well as negative among the homogenous and heterogeneous nucleotides, respectively (Figure 4E). The consequence of selection pressure on the CUB was observed in terms of correlation to be positive and highly significant between homogeneous nucleotides (A/T vs. A3/T3; G/C vs. G3/C3); however, the correlation value was observed to be negative and highly significant among heterogeneous nucleotides (A/T vs. G3/C3; G/C vs. A3/T3) and indicated that mutation pressure perhaps also has affected the CUB of genes involved in human autophagy. The results obtained showed that both pressures (mutation and natural selection) could have resulted in shaping the codon usage bias of human autophagy genes.

### 3.6. Correspondence Analysis

The values of RSCU obtained from the coding sequences of the genes involved in human autophagy were utilized to perform a correspondence analysis (or CoA). The two axes of the CoA, namely, axis 1 and axis 2 contributed 41.31% and 4.60% out of the complete variation (Figure 5). The distribution of codons closer to the axes suggested that the CUB of genes involved in human autophagy was influenced by the nucleotide constituents [52]. Furthermore, the scattering of some of the genes in a discrete fashion signified that other determinants such as natural selection perhaps affected the CUB of the genes [53].

### 3.7. Codon Context Analysis

The codon context examination (64 codon × 64 codons) was performed using Anaconda software to recognize the framework of the codon context of genes linked with human autophagy [38,50]. The codon context patterns were analyzed by generating a heatmap where we observed prominent variations in codon pair preferences (Figure 6). The clustered array among the gene relies on the mean matrix of residuals of the individual codon context. The 5′ codons are placed in rows, while the columns represent the 3′ codons.

### 3.8. Correlation between ENC and Protein Properties

A correlation study was performed to examine the effect of protein properties on the CUB of human autophagy genes. The correlation study among the ENC and protein characteristics such as protein stretch in terms of amino acid number, pI, GRAVY, and aromaticity was found to be highly significant (Figure 7). The regression coefficients of the ENC on pI (−0.715) and GRAVY (−25.23) were negative but positive on protein lengths (0.003) and Aromo (17.40). The results thus obtained indicate that GRAVY and pI influenced the CUB of genes involved in human autophagy in a positive manner.

### 3.9. Minimum Free Energy (mFE) of mRNA and Codon Usage Pattern of Human Autophagy Genes

A correlation analysis between the ENC and mFE was performed to understand the proportion of stability of mRNA for all the genes involved in human autophagy. For this purpose, the mFE values of mRNAs were calculated using the RNAfold web server available at http://rna.tbi.univie.ac.at/ accessed on 29 September 2022 [44,45]. Due to the longer mRNA sequence than the permitted limit of the server, the mFE of seven mRNAs of genes (*ATG2B, BIRC6, CFLAR, EIF2AK2, NCKAP1, NRG1*, and *WDFY3*) could not be calculated. Therefore, the analysis has been performed with the rest of the 217 autophagy gene mRNAs. The average mFEs of all these genes were found to be −1186.59 kcal/mol, with values fluctuating from −212.2 kcal/mol to −3155.12 kcal/mol (Appendix A). The negative sign is representative of energy being lost from the mRNA proceeding in an additionally stable mRNA configuration. The results were based on the analysis we performed, and we noted an extremely significant positive correlation (*p* < 0.01) among mFE and ENC, indicating that the release of mFE by the mRNA molecule could be related to the intensity of the CUB of autophagy genes (Figure 8). A further analysis of correlation was also carried out among mFE and GC constraints such as GC, GC1, GC2, and GC3, and we found a highly significant positive correlation (*p* < 0.01) among mFE and GC compositions at all the positions (Figure 8). These results suggested that the mFE of human autophagy gene mRNAs might be related to GC constraints, and that more liberation of free energy by mRNAs could perhaps be accompanied by high GC content.

## 4. Discussion

The bias in codon usage emerges from the uneven use of synonymous codons in fully developed mRNA transcripts, and an analysis of the CUB is regarded as a well-established method for retrieving significant genetic information and comprehending the mechanism of transformation at the molecular extent. After the genome sequencing of numerous organisms and that information being available online in various databases, and the advent of advanced bioinformatics tools, the study of patterns of the CUB became an interesting and valuable field of research over the past few years [54].

In this work, we studied the patterns of the CUB across 224 protein coding genes involved in human autophagy upon the impact of two major evolutionary forces, namely, mutation and natural selection [28]. Recently, research was made on patterns of the CUB in human tumor suppressor genes [55]; Y-linked genes [56]; and mitochondrial ATP genes in mammals, fishes, and aves [57], but no work has been reported in autophagy-related genes so far, which makes our research significant in providing molecular insights of genes linked with autophagy in humans. Analyzing our current study, the average ENC value for genes involved in human autophagy was found to be 45.05, indicating a low CUB. Similar studies on tumor suppressor genes reported a low CUB with an average ENC value of 48 [55], and the mean ENC value of Y-linked genes in humans was 50.33 indicating a low CUB in those genes [56]. Analyzing various compositions of nucleotides serves as a significant aspect for molding the bias of codon usage in a gene. The primary investigation of nucleotide composition in human autophagy genes led to the observation that G% was highest accompanied by A%, C%, and T%. The overall content of the GC nucleotide (52.23%) marginally surpassed the content of nucleotide AT (Figure 1). Related outcomes were also published beforehand in tumor suppressor genes with average GC% being 54.4 [55], and genes linked with anxiety with an average constituent of nucleotide GC being 54.76 [53].

The RSCU analysis of human autophagy genes revealed three over-represented codons (RSCU > 1.6) CTG, GTG and ATC, and ten under-represented codons (RSCU < 0.6) CGT, ACG, CAA, GTA, CCG, TTA, GCG, ATA, CTA and TCG (Figure 2). Most of the remaining codons were more (RSCU between 1–1.6) or less (RSCU between 0.6–1) frequently used codons. The RSCU analysis of tumor suppressor genes provided two codons that were over-represented, CTG and GTG, and 9 codons that were under-represented, GTA, TCG, CCG, CTA, TTA, etc., thus further supporting our results [55]. Additionally, the RSCU analysis of pancreatic cancer-associated genes provided two codons that were over-represented: CTG and GTG, and 10 codons that were under-represented, TCG, TTA, CTA, CCG, CAA, etc., thus further supporting our results [58]. Overall, the RSCU analysis inferred low bias in terms of the usage of codons in human autophagy genes.

In this study, the CUB of genes involved in human autophagy was found to be influenced by the GC ending codons (Figure 3). The ENC is observed to be a non-directional estimation of the CUB; therefore, the higher the values of ENC, the much lower the CUB, and vice-versa. As we observed negative regression coefficients of ENC upon GC contents, it indicated the positive influence of GC on the CUB in human autophagy genes [50]. The significant negative correlation coefficients of the ENC against GC constraints further support the dependence of the CUB on GC ending codons. Similar results have earlier been observed where GC ending codons dominate the CUB of human tumor suppressor genes [55]. Our further interpretation on studying the correlation of codon usage with GC3 revealed that the codon usage of human autophagy genes was positively correlated to GC3 except for AGG and TTG (Figure 3M). On the other hand, the usage of AT ending codons was negatively correlated to GC3. Earlier work by Chakraborty et al. (2019) on human TP63 gene isoforms also reported the similar GC3 correlation with GC ending codon usage [59]. The CUB in human Y-linked genes also reported positive correlations between GC3 and all the GC ending codons [56]. Overall, these analyses suggest a strong influence of GC ending codons on the CUB of human autophagy genes.

The CUB of genes is influenced by two important evolutionary forces: the mutational pressure and natural selection. Parity rule 2 bias plots were analyzed to investigate the effect of these evolutionary forces by examining the relationship between GC and AT bias (Figure 4A). In our analysis, we observed a variation between GC and AT bias encouraging that the pair of evolutionary forces governed the CUB in autophagy genes, further supporting the results reported in human tumor suppressor genes [55] as well as Y-linked genes in humans [56]. Further, the neutrality plot analysis revealed a significant positive correlation between GC12 and GC3, and a wide distribution of GC3 values. All the results from our study directed towards the two principal forces, one of them being natural selection and the other being mutation pressure, perhaps affected the CUB of autophagy genes in humans, and these overall results also corresponded with human tumor suppressing genes [55]. To further validate this observation, a correlation analysis among the entire composition of nucleotides and the composition of nucleotides at the third position of codons was performed which showed significant positive values of the correlation coefficient between homogeneous bases, and negative values between heterogeneous bases (Figure 4E). The results thus obtained indicated that both primary forces of nature, mutation pressure and natural selection, shaped the CUB of genes involved in human autophagy. Similar works performed on human mitochondrial genes supported our results, revealing both the key forces of evolution governed the patterns of the CUB in those genes as well [60]. Moreover, it has been strongly suggested that the CUB of autophagy-related gene *ATG13* across different species exhibited a disproportionate influence of mutational pressure and natural selection in its evolution [24]. Therefore, a comprehensive study on the complete repertoire of autophagy genes across multiple species would provide a deeper understanding.

The CoA showed the distribution of most of the points closer to axes indicating the nucleotide composition and mutational pressure influence on the CUB, whereas, some genes were found to be scattered in a discontinuous distribution that suggests natural selection was shaping the CUB of autophagy genes [52,53]. Additionally, we found that protein properties such as GRAVY and pI had a positive influence on the CUB of autophagy genes. Nath Choudhury et al. 2017, also represented the positive correlation between ENC and GRAVY in human Y-linked genes [56].

Since mRNA plays a significant role in translation, factors influencing mRNA affect the process of protein synthesis [61]. A low mFE value of coding sequences signifies a weak folding of mRNA and contributes to generating more proteins than those bearing high mFE values [61]. Our analysis displayed a mean mFE value of −1186.59 kcal/mol with a range from −212.2 kcal/mol to −3155.12 kcal/mol, suggesting that genes with the lowest values might have more elevated rates of translation than those with higher values of mFE. The correlation among mFE and ENC was found to be significantly positive suggesting that the extent of the CUB is associated with the mFE of mRNA. The value of correlation perceived among mFE and GC compositions also appeared to be significantly positive, conveying that GC content affects mRNA stability, which might play a significant function in the determination of gene expression.

## 5. Conclusions

Established on the profound evidence of nucleotide compositions and codon usage studies, it can be concluded that the human autophagy genes possess a higher GC content and show low CUB, suggesting a high variability in synonymous codon usage. Despite numerous factors indulged with influencing the codon usage bias occurring in a gene, the two significant factors that contribute greatly to framing the codon usage of human autophagy genes are the natural selection and mutation pressure. These data reveal the unique perception of patterns of codon usage and their nucleotide compositions in genes involved in human autophagy. Elucidating the patterns of codon usage in human autophagy genes provides novel insights into their genetic characteristics, the molecular evolution of these genes, and contributes to molecular details of the autophagic pathway in humans. Since the evolution and conservation of human autophagy genes still require profound clarity, our analysis on the composition of nucleotide, RSCU, and determining the forces of nature governing CUB may contribute to a crucial investigation at the genetic level and its evolutionary trend in human autophagy genes.

## Figures and Tables

**Figure 1 cells-11-03203-f001:**
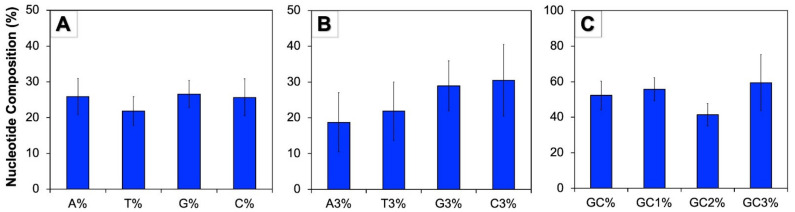
Nucleotide composition of coding sequences of human autophagy genes. (**A**): Average composition of individual bases among 224 human autophagy genes; (**B**): A, T, G, and C frequency at the third position of the codons; (**C**): Distribution of overall GC content and GC content at the first, second, and third codon position.

**Figure 2 cells-11-03203-f002:**
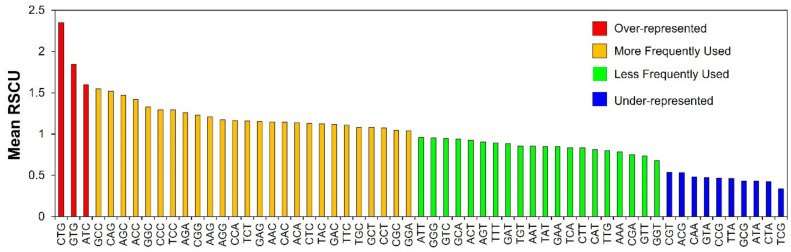
RSCU values for individual synonymous codons of human autophagy genes. The figure depicts the over-represented (RSCU > 1.6), under-represented (RSCU < 0.6), more frequently used (RSCU between 1 and 1.6), and less frequently used (RSCU between 0.6 and 1) codons.

**Figure 3 cells-11-03203-f003:**
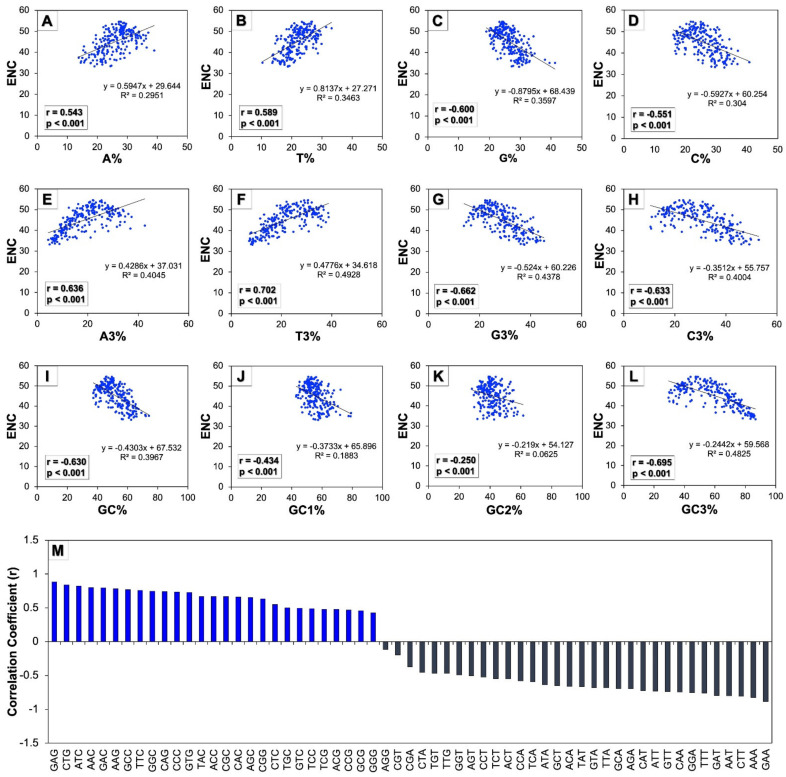
Regression analysis among ENC and various nucleotide composition of human autophagy genes. Scattered regression plots between ENC and individual nucleotide composition (**A**–**D**), between ENC and nucleotides at third codon position (**E**–**H**), and between ENC and GC contents (**I**–**L**) were analyzed. The insets in all the regression plots represent their corresponding correlation coefficients and *p* values. (**M**) represents the correlation between codon usage and GC3 across all 224 genes involved in human autophagy.

**Figure 4 cells-11-03203-f004:**
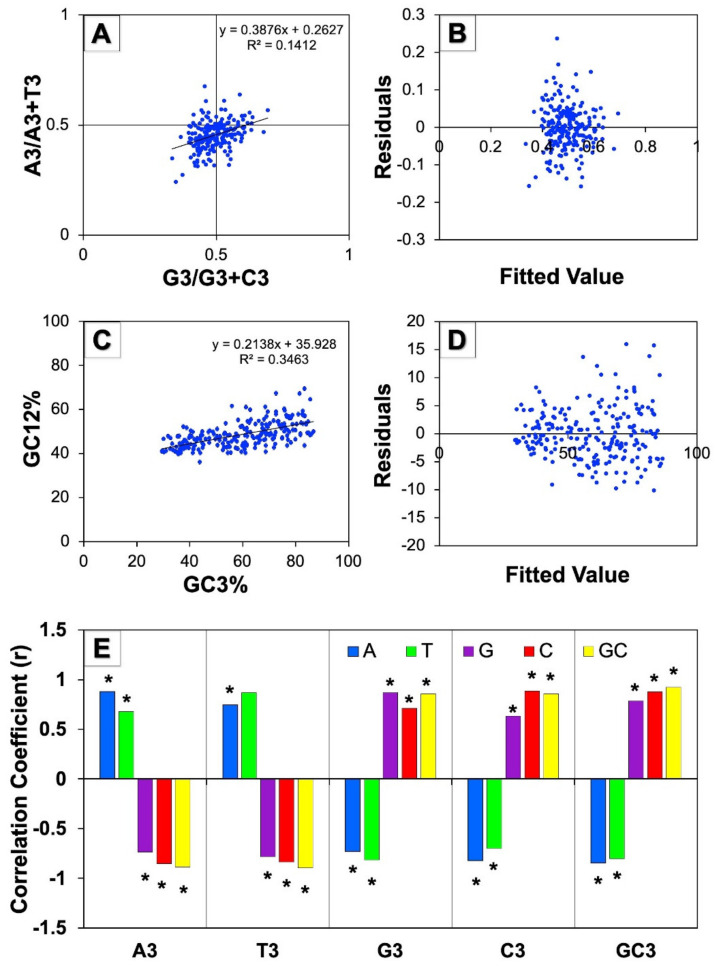
Influence of evolutionary forces on CUB of human autophagy genes. (**A**) represents the parity rule 2 bias analysis of human autophagy genes showing disproportionate distribution of GC and AT, while (**B**) is the residual plot for analyzing this regression model; (**C**) represents the neutrality plot analysis between GC12 (average of GC1 and GC2) and GC3 for all the 224 human autophagy genes, while (**D**) is the corresponding residual plot. (**E**) represents the correlation between overall nucleotide composition and nucleotide composition at the third codon position (* indicates *p* < 0.01).

**Figure 5 cells-11-03203-f005:**
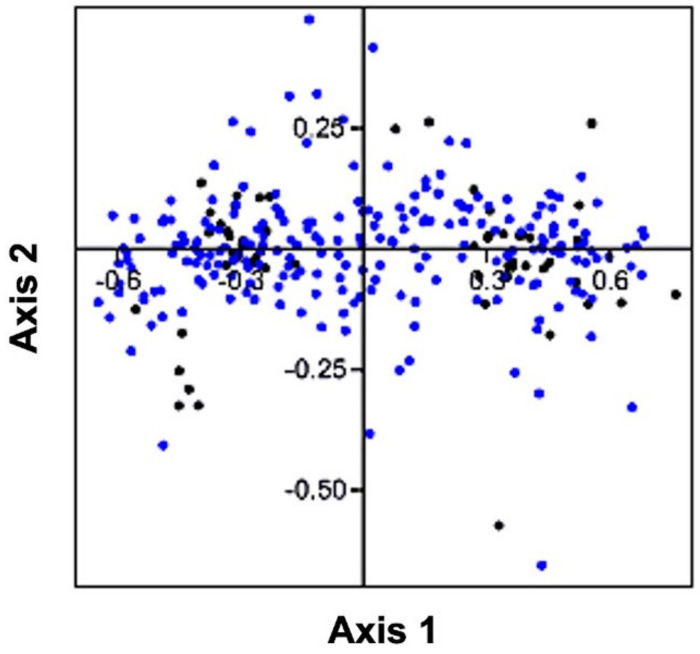
Correspondence analysis (CoA) of human autophagy genes. The CoA is represented by blue and black colored dots designating the genes and their codons, respectively.

**Figure 6 cells-11-03203-f006:**
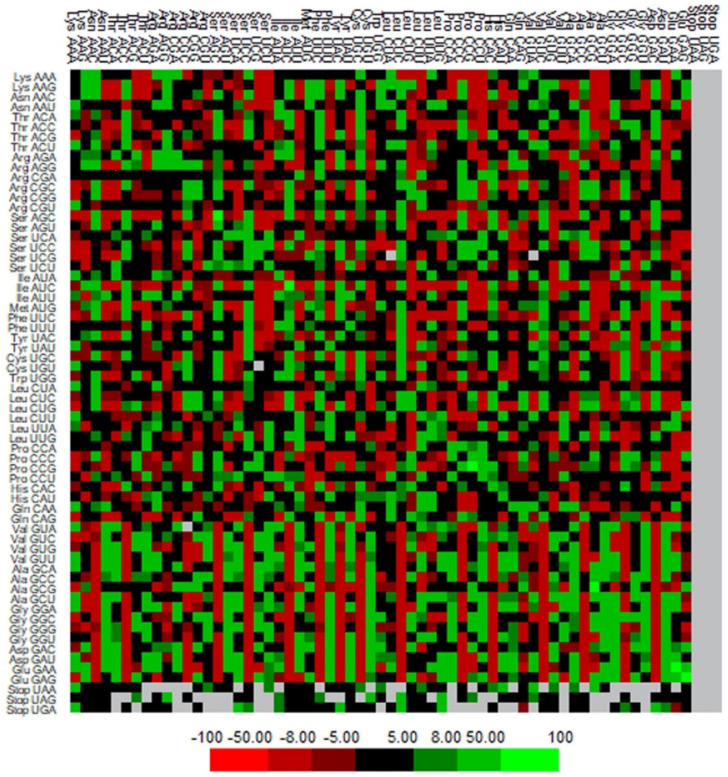
Codon context of 224 human autophagy genes. Highest and lowest numbers of codon contexts are represented by green and red colors, respectively.

**Figure 7 cells-11-03203-f007:**
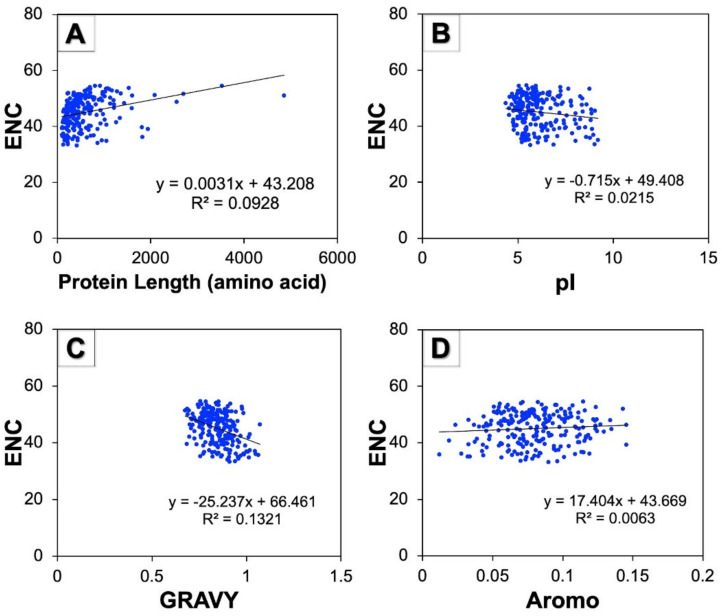
Regression analysis among ENC and various protein properties of human autophagy genes. (**A**–**D**) represent the scattered regression plots between ENC and protein length, pI, GRAVY, and Aromo, respectively.

**Figure 8 cells-11-03203-f008:**
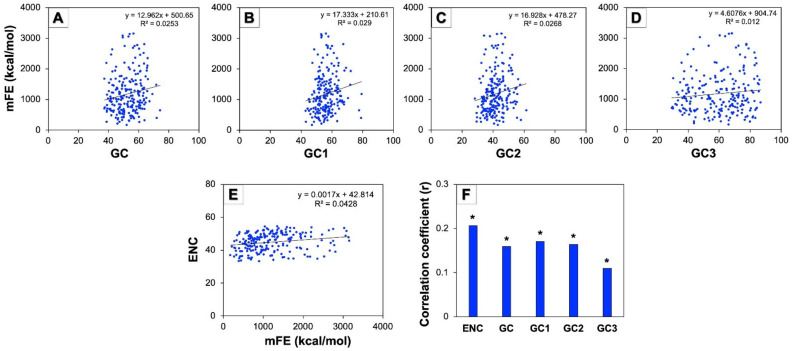
Role of mFE of genes involved in human autophagy on CUB. (**A**–**D**) represent the regression plots between mFE and GC contents in all genes. (**E**) represents the regression plot between ENC and mFE. (**F**) represents the distribution of the correlation coefficient of mFE against ENC and various GC contents such as overall GC, GC1, GC2, and GC3. For these analyses, absolute values of mFE were considered (* represents *p* < 0.01).

## Data Availability

The data presented in this study are available in this article.

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
