# Peer review of "Analysis of the Compositional Features and Codon Usage Pattern of Genes Involved in Human Autophagy"

_cells, 2022, doi:10.3390/cells11203203_

Round 1
Reviewer 1 Report (Previous Reviewer 2)
Many thanks to the authors for trying to address the comments in the previous round of review. Even though the work looks more solid now, there are still some issues that need to be resolved.
+ Major comments
Some results presented seem to be superficially explained and not fully supported by the underlying data.
For example the first two sections in Results are titled:
"Human autophagy genes contain higher GC content"
"Human autophagy genes show low codon usage bias"
Even though the above statements might be valid, they cannot be supported by the analysis presented in the absence of any reference/control points. How much low/high values should be for these statements to be valid? Is there any theory supporting particular thresholds for GC content or codon usage bias? A random set of human genes (unrelated to autophagy) could be used to provide some baseline estimates and form the basis for statistically evaluating the claims made by the authors.
Similarly, the RSCU levels chosen in section 3.3. to group genes are arbitrary: why were the particular values chosen?
+Minor comments/typos
L124. "exact multitude of three nucleotides": here the authors may want to say that the length of the coding sequence is a multiple of three?
L53 and L154. The mathematical notation wrt parity plots needs to be revised. Correct forms: "A3/(A3+T3)" and "G3/(G3+C3)".
L172. A reference to the ' “Past” software ' is missing here.
L177. The version of the ANACONDA software used in ref [37] is V2.0. If this was the version used in this work then it should be clearly stated as such. Otherwise, the original papers describing the method should be preferably cited (see below)
Moura G, Pinheiro M, Silva R, Miranda I, Afreixo V, et al. (2005) Comparative context analysis of codon pairs on an ORFeome scale. Genome Biol 6: R28.
Pinheiro M, Afreixo V, Moura G, Freitas A, Santos MA, et al. (2006) Statistical, computational and visualization methodologies to unveil gene primary structure features. Methods Inf Med 45: 163–168.
L185. I am not sure whether this is the most appropriate citation for CodonW, however I believe the authors should cite the PhD thesis of the developer (John Peden, http://codonw.sourceforge.net/JohnPedenThesisPressOpt_water.pdf) or some of the publications that resulted from this thesis.
L205. Please give a citation to CAIcal
Puigbo P, Bravo IG and Garcia-Vallve S. (2008) CAIcal: a combined set of tools to assess codon usage adaptation. Biology Direct, 3:38.
and check about appropriate citations for all other software used in this work.
Author Response
Thank you for the valuable comments. In this version of the manuscript, all attempts have been made to update the manuscript in light of your comments. All changes in the revised manuscript have been highlighted. A point-by-point response to the comments are given in the attached file.
Many thanks to the authors for trying to address the comments in the previous round of review. Even though the work looks more solid now, there are still some issues that need to be resolved.
Response: Thank you for the comment. In this version of the manuscript, all attempts have been made to update the manuscript in light of your comments. All changes in the revised manuscript have been highlighted. A point-by-point response to the comments are given below.
+ Major comments
Some results presented seem to be superficially explained and not fully supported by the underlying data.
For example the first two sections in Results are titled:
"Human autophagy genes contain higher GC content"
"Human autophagy genes show low codon usage bias"
Even though the above statements might be valid, they cannot be supported by the analysis presented in the absence of any reference/control points. How much low/high values should be for these statements to be valid? Is there any theory supporting particular thresholds for GC content or codon usage bias? A random set of human genes (unrelated to autophagy) could be used to provide some baseline estimates and form the basis for statistically evaluating the claims made by the authors.
Response: The localization of the genes in GC regions of the genome has been well documented and reported across many species since the human genome project (10.1016/s0378-1119(02)01038-7; 10.1186/s12864-019-6131-1). So, we understand and acknowledge the fact that coding regions of the genome are expected to have a higher GC content. Since the present study is primarily focussed on codon bias, the genome content of the studied genes were looked at to provide context. We have not tried to emphasize in any manner as it is a significant finding but merely stated the observed data and kept it as reference for analysing the codon bias.
Regarding the codon usage bias, Wright (1990) (doi: 10.1016/0378-1119(90)90491-9) gave the concept of effective number of codons (ENC) to recognize the bias in the use of synonymous codons. The value of ENC ranges from 20 to 61. An ENC value of 20 indicates the amino acid is extremely biased, that means the amino acid is encoded by only single codon despite having different synonymous codons. In contrast, an ENC value of 61 indicates the amino acid shows no bias in codon usage which means all the synonymous codons are used in uniform manner. Generally, a genome is considered as strongly biased if the observed ENC value is less than 35. Further, if the value of ENC is greater than 40, the codon usage bias is regarded as low (Wright, 1990).
Our analysis also follows the same criteria as has been mentioned in Section 3.2 (L242-247). Supporting references have been added
Similarly, the RSCU levels chosen in section 3.3. to group genes are arbitrary: why were the particular values chosen?
Response: RSCU is a CUB index evaluated as the ratio of the observed frequency of a codon to its expected frequency out of all synonymous codons for a particular amino acid, multiplied by the degeneracy level. An RSCU value of 1 indicates equal usage of all synonymous codons, while a value greater than 1 indicates that a particular codon is favored (Deb et al., 2021, https://doi.org/10.1007/s00705-020-04890-2). RSCU values are generally categorized into four groups: (a) Over-represented with RSCU values above 1.6, (b) Under-represented with RSCU values below 0.6, (c) More frequently used with RSCU between 1-1.6 and (d) Less frequently used with RSCU between 0.6 and 1 (Behura and Severson, 2012, https://doi.org/10.1371/journal.pone.0043111; Wong et al., 2022, doi: https://doi.org/10.3390/ijms23137441; Deb et al., 2021, doi: https://doi.org/10.1007/s00705-020-04890-2). In our study, we have categorised the RSCU values based on the same principle.
+Minor comments/typos
L124. "exact multitude of three nucleotides": here the authors may want to say that the length of the coding sequence is a multiple of three?
Response: With the phrase it was meant that the total number of nucleotides in selected coding sequences (CDS) was a multiple of three. In the revised manuscript it has been replaced as “exact multiples of three nucleotides” for better comprehension.
L53 and L154. The mathematical notation wrt parity plots needs to be revised. Correct forms: "A3/(A3+T3)" and "G3/(G3+C3)".
Response: Thank you for pointing out the inadvertent typographical error. It has been updated in the revised manuscript.
L172. A reference to the ' “Past” software ' is missing here.
Reference: The reference has been added in the revised manuscript (L174).
L177. The version of the ANACONDA software used in ref [37] is V2.0. If this was the version used in this work then it should be clearly stated as such. Otherwise, the original papers describing the method should be preferably cited (see below).
Moura G, Pinheiro M, Silva R, Miranda I, Afreixo V, et al. (2005) Comparative context analysis of codon pairs on an ORFeome scale. Genome Biol 6: R28.
Pinheiro M, Afreixo V, Moura G, Freitas A, Santos MA, et al. (2006) Statistical, computational and visualization methodologies to unveil gene primary structure features. Methods Inf Med 45: 163–168.
Reference: ANACONDA V2.0 has been used in this study and it has been updated in the text (L177).
L185. I am not sure whether this is the most appropriate citation for CodonW, however I believe the authors should cite the PhD thesis of the developer (John Peden, http://codonw.sourceforge.net/JohnPedenThesisPressOpt_water.pdf) or some of the publications that resulted from this thesis.
Reference: The reference has been added in the revised manuscript (L177).
L205. Please give a citation to CAIcal
Puigbo P, Bravo IG and Garcia-Vallve S. (2008) CAIcal: a combined set of tools to assess codon usage adaptation. Biology Direct, 3:38.
Reference: The reference has been added in the revised manuscript (L206).
and check about appropriate citations for all other software used in this work.
Reference: All the reference has been cross checked and updated in the revised manuscript.
Reviewer 2 Report (New Reviewer)
Jamil et al performed bioinformatic analyses of the composition of the human autophagy-related genes. Their studies used the same approaches many others have extensively applied, for e.g to anxiety-related genes (https://pubmed.ncbi.nlm.nih.gov/32813237/). Similar conclusions were reached as in the other reports, yet the biological significance of such analyses were not demonstrated. At the end of the day, what do we learn from these analyses, except from knowing which codons were presented at higher/lower probabilities than others in those genes?
Author Response
Jamil et al performed bioinformatic analyses of the composition of the human autophagy-related genes. Their studies used the same approaches many others have extensively applied, for e.g to anxiety-related genes (https://pubmed.ncbi.nlm.nih.gov/32813237/). Similar conclusions were reached as in the other reports, yet the biological significance of such analyses were not demonstrated. At the end of the day, what do we learn from these analyses, except from knowing which codons were presented at higher/lower probabilities than others in those genes?
Response: There have been various tools in use for the analysis and exploration of genomes in recent times. These include various predictive studies in terms of genomics/proteomics, genome/sequence annotation tools, polymorphism studies, repeat sequences and so on. Similarly, codon bias has been one of the approaches to study and understand any preferential usage of codons in a genome/organism as a whole or on functionally relevant set of genes. This helps elucidate the comparative and functional genomics of the genes. Present study is an attempt to elucidate the patterns of codon usage in human autophagy genes wherein, the patterns are indicative of clear bias in codon representation which in turn is significant in defining the evolution of those genes.
Round 2
Reviewer 1 Report (Previous Reviewer 2)
I thank the authors for the revised version of this work.
No further comments for revisions on my side.
Author Response
Please see the attachment

Reviewer 2 Report (New Reviewer)
Authors argue that their analyses were important as they shed light on the evolution of certain human genes. They should elaborate on 2 points:
1. From their analyses, what did we learn about the evolution of this group human genes? Did authors perform any analyses on related genes in other organisms to make a statement in their responses to my critique?
2. Why they think understanding the evolution of these genes is important? Is this helpful to understand related human diseases or guide therapeutic development?
In short, again, what we can learn from such analyses about human autophagy?
Author Response
Please see the attachment

This manuscript is a resubmission of an earlier submission. The following is a list of the peer review reports and author responses from that submission.
Round 1
Reviewer 1 Report
This is an example of a paper that is correctly executed, clearly written and organised, but it is inconclusive with respect to the scientific problem that is alluded to. Of course, this is not an exception, looking at the standard literarure in the field. But this is not a justification. A scientific paper should point either to a scientific well defined problem or, in the case of observational studies, should give a clear set of reference value, in order to establish the elements of a classification. The defect of this work is that at the end of reading one is not able to answer basic questions such as: are ATG genes under a darwinian selection that is bigger than in the bacterial essential genes or not?
It seems to be just an exploratory collection of CUB signals, that are correctly and neatly exposed. But the big question remains: what for? and, most importantly: So What? What one has learnt on autophagy after reading this work? The point is that there are not comparisons, with respect to standard reference sets of genes.
Apparently, the declared object of this study is to establish the relative role that mutational pressure and darwinian selection on the CUB of ATG genes. In the discussion there is reference to other similar studies on: human tutor suppressor genes, Y-linked genes, mitochondrial ATP genes in mammals, fishes and aves.
Though a wealth of other set of genes have been explored, in the literature, for CUB, using the same set of observables the authors use here, I think that the authors, in the discussion, should systematically and conclusively compare the relative role of mutational pressure and natural selection at least in the sets of gene they refer to in references 48,49,50.
The paper, in my opinion does not require any other improvement, but a careful comparison with similar computational CUB assessment in other significant group of genes.
Author Response
Response to Reviewer Comments
This is an example of a paper that is correctly executed, clearly written and organised, but it is inconclusive with respect to the scientific problem that is alluded to. Of course, this is not an exception, looking at the standard literarure in the field. But this is not a justification. A scientific paper should point either to a scientific well defined problem or, in the case of observational studies, should give a clear set of reference value, in order to establish the elements of a classification. The defect of this work is that at the end of reading one is not able to answer basic questions such as: are ATG genes under a darwinian selection that is bigger than in the bacterial essential genes or not?
Response: We thank the reviewer for appreciating the manuscript. The findings of this study suggest the influence of both natural selection and mutational pressure governing codon usage bias of human ATG genes. However, to compare the same with bacterial genes is beyond the scope of the study.
It seems to be just an exploratory collection of CUB signals, that are correctly and neatly exposed. But the big question remains: what for? and, most importantly: So What? What one has learnt on autophagy after reading this work? The point is that there are not comparisons, with respect to standard reference sets of genes.
Response: Manuscript has been revised in light of the kind comments of the expert reviewer. Significance of studying CUB of human ATG genes has been clearly described in the revised manuscript (Introduction, discussion and conclusion sections). Moreover, our findings have been compared with previously reported CUB in human tumor suppressor genes and Y-linked genes in the revised version of the manuscript.
Apparently, the declared object of this study is to establish the relative role that mutational pressure and darwinian selection on the CUB of ATG genes. In the discussion there is reference to other similar studies on: human tutor suppressor genes, Y-linked genes, mitochondrial ATP genes in mammals, fishes and aves.
Response: In the discussion section, the findings of this study of the influencing forces on the CUB of ATG genes have been compared with previously reported CUB in similar human gene sets: tumor suppressor genes and Y-linked genes in the revised version of the manuscript.
Though a wealth of other set of genes have been explored, in the literature, for CUB, using the same set of observables the authors use here, I think that the authors, in the discussion, should systematically and conclusively compare the relative role of mutational pressure and natural selection at least in the sets of gene they refer to in references 48,49,50.
Response: We appreciate the reviewer’s suggestion. Accordingly, our findings on the relative roles of mutational pressure and natural selection on the CUB of ATG genes have been compared with previously reported CUB in similar human gene sets: tumor suppressor genes and Y-linked genes in the revised version of the manuscript.
The paper, in my opinion does not require any other improvement, but a careful comparison with similar computational CUB assessment in other significant group of genes.
Response: We thank the reviewer for the appreciation. All care has been taken and the manuscript revised in light of the kind comments of the reviewers.
Reviewer 2 Report
In this work, Jamil and colleagues present an analysis of compositional features and codon usage patterns of human autophagy-related genes.
The authors collected 36 nucleotide sequences of genes involved in autophagy in humans and used existing software tools for the above mentioned analysis. They interpret their results by concluding that codon usage in this set of genes is shaped by both natural selection and mutation pressure.
The topic of this work is interesting. However, there are a few methodological points that raise concerns on the validity of the conclusions. Several minor points - ranging from simple typographical errors to weakly supported conclusions - are also found in the manuscript however they are not highlighted below. In a few instances, the exact analyses performed and the validity of statistical handling of the results need to be reexamined or presented in more detail.
Major points
- A central point to this paper is the choice of genes to be analysed. The authors used the HGNC resource to retrieve autopahgy-ralted genes in human: even though they do not disclose the exact method they used to collect these genes, I could verify that they performed a simple keyword search using the "ATG" keyword. As a result, three of the genes analysed in this work (namely ST3GAL5, PNPLA2 and NONO) have no obvious functional link to autophagy: they are only retrieved from HGNC because some of their gene name aliases contains somewhere the characters 'ATG'. For example, NONO is also known as 'ATGCAAAT binding protein'. I would suggest that if the authors want to perform a meaningful analysis of genes participating in autophagic processes, they either consult a recent comprehensive review (for example the review by the group of Noboru Mizushima https://doi.org/10.1242/jcs.233742) or use data from a well annotated sequence database. UniProt/SwissProt entries annotated to be involved in autophagy or the Human Autophagy database (http://autophagy.lu/) would make excellent choices.
- Several of the genes involved in autophagy come in many alternative transcripts, often leading to the production of different isoforms. The authors seem to arbitrarily choose one of these transcripts for their analyses, however it is not mentioned how they choose which transcript they analyse.
- For the free energy calculations, it is the complete mRNA sequences that need to be analysed and not only the coding sequences. In any other case the results are devoid of any true biological significance.
Author Response
Response to Reviewer Comments
In this work, Jamil and colleagues present an analysis of compositional features and codon usage patterns of human autophagy-related genes.
The authors collected 36 nucleotide sequences of genes involved in autophagy in humans and used existing software tools for the above mentioned analysis. They interpret their results by concluding that codon usage in this set of genes is shaped by both natural selection and mutation pressure.
The topic of this work is interesting. However, there are a few methodological points that raise concerns on the validity of the conclusions. Several minor points - ranging from simple typographical errors to weakly supported conclusions - are also found in the manuscript however they are not highlighted below. In a few instances, the exact analyses performed and the validity of statistical handling of the results need to be reexamined or presented in more detail.
Response: We greatly appreciate and thank the reviewer for evaluating the manuscript and providing suggestions for further improvement. All attempts have been made to revise the manuscript, taking into consideration the kind comments and suggestions.
Major points
- A central point to this paper is the choice of genes to be analysed. The authors used the HGNC resource to retrieve autopahgy-ralted genes in human: even though they do not disclose the exact method they used to collect these genes, I could verify that they performed a simple keyword search using the "ATG" keyword. As a result, three of the genes analysed in this work (namely ST3GAL5, PNPLA2 and NONO) have no obvious functional link to autophagy: they are only retrieved from HGNC because some of their gene name aliases contains somewhere the characters 'ATG'. For example, NONO is also known as 'ATGCAAAT binding protein'. I would suggest that if the authors want to perform a meaningful analysis of genes participating in autophagic processes, they either consult a recent comprehensive review (for example the review by the group of Noboru Mizushima https://doi.org/10.1242/jcs.233742) or use data from a well annotated sequence database. UniProt/SwissProt entries annotated to be involved in autophagy or the Human Autophagy database (http://autophagy.lu/) would make excellent choices.
Response: We greatly appreciate the expert comment. In the revised version of the manuscript, the stepwise method for selection of human ATG genes has been described in Methods Section 2.1. Only the genes involved in human autophagy process have been considered and their functions in human autophagy has been added in Supplementary Table S1.
The inadvertently incorporated data of the said three genes have been eliminated and all the analysis performed again with the remaining 33 ATG genes. Accordingly, all the figures and relevant tables have been updated.
- Several of the genes involved in autophagy come in many alternative transcripts, often leading to the production of different isoforms. The authors seem to arbitrarily choose one of these transcripts for their analyses, however it is not mentioned how they choose which transcript they analyse.
Response: For this study, the transcript details given in HGNC database for each ATG gene were considered. We have incorporated the selection procedure for the transcripts in Methods Section 2.1, in the revised manuscript.
- For the free energy calculations, it is the complete mRNA sequences that need to be analysed and not only the coding sequences. In any other case the results are devoid of any true biological significance.
Response: We are thankful for the scholarly comment. The complete sequences of mRNA for all the human ATG genes were considered for calculating mFE and re-analyzed in the revised manuscript. Accordingly, the revised figure is presented as Figure 8.
Round 2
Reviewer 1 Report
I have considered the answers to my previous remarks and I re-read with interest the revised manuscript. I think that the authors were successful in making their work more focussed and to put it in a more clear methodological perspective. This is why I substantially shifted the overall rating of this contribution from average to high. We have now a piece of sound work that contributes to the evolutionary study of codon bias.
Reviewer 2 Report
My main concern regarding the selection of genes to be analyzed in this work still remains unresolved. I agree that the authors deleted the 3 irrelevant genes initially included in the analysis, however the rationale of the selection of the dataset is not substantiated.
The list of genes described here in (ATG genes) are linked by name (autophagy-related) and are no more coherent as a group as any other subset of genes performing functions related to autophagy. Thus, the selection of genes (which is crucial for setting any biological question to be subsequently addressed) is not convincing.
For example, by performing a simple UniProt query
https://www.uniprot.org/uniprot/?query=goa%3A%28%22autophagy+%5B6914%5D%22%29+organism%3A%22Homo+sapiens+%28Human%29+%5B9606%5D%22&sort=score
to retrieve human entries with annotated functionality in autophagy 643 entries are retrieved, with 282 of them being Reviewed entries. Many of these proteins are not assigned "ATG" names (I suspect for historical reasons only) but are players in this key eukaryotic catabolic process. I will only list a few very well known cases here: Sequestosome-1 (Q13501), Reticulophagy regulator 3 (Q86VR2), WD repeat and FYVE domain-containing protein 3 (a.k.a. "Alfy", Q8IZQ1).
Alternatively, I had suggested the use of the Human Autophagy database (http://autophagy.lu/) where the authors could again obtain a manually curated list of proteins participating in autophagic processes, or recent reviews on the topic.